# TOWARDS THE LIMIT OF NETWORK QUANTIZATION

**Yoojin Choi, Mostafa El-Khamy, and Jungwon Lee**
Samsung US R&D Center, San Diego, CA 92121, USA
`{yoojin.c,mostafa.e,jungwon2.lee}@samsung.com`

## ABSTRACT

Network quantization is one of network compression techniques to reduce the redundancy of deep neural networks. It reduces the number of distinct network parameter values by quantization in order to save the storage for them. In this paper, we design network quantization schemes that minimize the performance loss due to quantization given a compression ratio constraint. We analyze the quantitative relation of quantization errors to the neural network loss function and identify that the Hessian-weighted distortion measure is locally the right objective function for the optimization of network quantization. As a result, Hessian-weighted k-means clustering is proposed for clustering network parameters to quantize. When optimal variable-length binary codes, e.g., Huffman codes, are employed for further compression, we derive that the network quantization problem can be related to the entropy-constrained scalar quantization (ECSQ) problem in information theory and consequently propose two solutions of ECSQ for network quantization, i.e., uniform quantization and an iterative solution similar to Lloyd's algorithm. Finally, using the simple uniform quantization followed by Huffman coding, we show from our experiments that the compression ratios of 51.25, 22.17 and 40.65 are achievable for LeNet, 32-layer ResNet and AlexNet, respectively.

## 1 INTRODUCTION

Deep neural networks have emerged to be the state-of-the-art in the field of machine learning for image classification, object detection, speech recognition, natural language processing, and machine translation (LeCun et al., 2015). The substantial progress of neural networks however comes with high cost of computations and hardware resources resulting from a large number of parameters. For example, Krizhevsky et al. (2012) came up with a deep convolutional neural network consisting of 61 million parameters and won the ImageNet competition in 2012. It is followed by deeper neural networks with even larger numbers of parameters, e.g., Simonyan & Zisserman (2014).

The large sizes of deep neural networks make it difficult to deploy them on resource-limited devices, e.g., mobile or portable devices, and network compression is of great interest in recent years to reduce computational cost and memory requirements for deep neural networks. Our interest in this paper is mainly on curtailing the size of the storage (memory) for network parameters (weights and biases). In particular, we focus on the network size compression by reducing the number of distinct network parameters by quantization.

Besides network quantization, network pruning has been studied for network compression to remove redundant parameters permanently from neural networks (Mozer & Smolensky, 1989; LeCun et al., 1989; Hassibi & Stork, 1993; Han et al., 2015b; Lebedev & Lempitsky, 2016; Wen et al., 2016). Matrix/tensor factorization and low-rank approximation have been investigated as well to find more efficient representations of neural networks with a smaller number of parameters and consequently to save computations (Sainath et al., 2013; Xue et al., 2013; Jaderberg et al., 2014; Lebedev et al., 2014; Yang et al., 2015; Liu et al., 2015; Kim et al., 2015; Tai et al., 2015; Novikov et al., 2015). Moreover, similar to network quantization, low-precision network implementation has been examined in Vanhoucke et al. (2011); Courbariaux et al. (2014); Anwar et al. (2015); Gupta et al. (2015); Lin et al. (2015a). Some extremes of low-precision neural networks consisting of binary or ternary parameters can be found in Courbariaux et al. (2015); Lin et al. (2015b); Rastegari et al. (2016). We note that these are different types of network compression techniques, which can be employed on top of each other.

The most related work to our investigation in this paper can be found in Gong et al. (2014); Han et al. (2015a), where a conventional quantization method using k-means clustering is employed for network quantization. This conventional approach however is proposed with little consideration for the impact of quantization errors on the neural network performance loss and no effort to optimize the quantization procedure for a given compression ratio constraint. In this paper, we reveal the suboptimality of this conventional method and newly design quantization schemes for neural networks. In particular, we formulate an optimization problem to minimize the network performance loss due to quantization given a compression ratio constraint and find efficient quantization methods for neural networks.

The main contribution of the paper can be summarized as follows:

- It is derived that the performance loss due to quantization in neural networks can be quantified approximately by the Hessian-weighted distortion measure. Then, Hessian-weighted k-means clustering is proposed for network quantization to minimize the performance loss.

- It is identified that the optimization problem for network quantization provided a compression ratio constraint can be reduced to an entropy-constrained scalar quantization (ECSQ) problem when optimal variable-length binary coding is employed after quantization. Two efficient heuristic solutions for ECSQ are proposed for network quantization, i.e., uniform quantization and an iterative solution similar to Lloyd's algorithm.

- As an alternative of Hessian, it is proposed to utilize some function (e.g., square root) of the second moment estimates of gradients when the Adam (Kingma & Ba, 2014) stochastic gradient decent (SGD) optimizer is used in training. The advantage of using this alternative is that it is computed while training and can be obtained at the end of training at no additional cost.

- It is shown how the proposed network quantization schemes can be applied for quantizing network parameters of all layers together at once, rather than layer-by-layer network quantization in Gong et al. (2014); Han et al. (2015a). This follows from our investigation that Hessian-weighting can handle the different impact of quantization errors properly not only within layers but also across layers. Moreover, quantizing network parameters of all layers together, one can even avoid layer-by-layer compression rate optimization.

The rest of the paper is organized as follows. In Section 2, we define the network quantization problem and review the conventional quantization method using k-means clustering. Section 3 discusses Hessian-weighted network quantization. Our entropy-constrained network quantization schemes follow in Section 4. Finally, experiment results and conclusion can be found in Section 5 and Section 6, respectively.

## 2 NETWORK QUANTIZATION

We consider a neural network that is already trained, pruned if employed and fine-tuned before quantization. If no network pruning is employed, all parameters in a network are subject to quantization. For pruned networks, our focus is on quantization of unpruned parameters.

The goal of network quantization is to quantize (unpruned) network parameters in order to reduce the size of the storage for them while minimizing the performance degradation due to quantization. For network quantization, network parameters are grouped into clusters. Parameters in the same cluster share their quantized value, which is the representative value (i.e., cluster center) of the cluster they belong to. After quantization, lossless binary coding follows to encode quantized parameters into binary codewords to store instead of actual parameter values. Either fixed-length binary coding or variable-length binary coding, e.g., Huffman coding, can be employed to this end.

### 2.1 COMPRESSION RATIO

Suppose that we have total $N$ parameters in a neural network. Before quantization, each parameter is assumed to be of $b$ bits. For quantization, we partition the network parameters into $k$ clusters. Let $\mathcal{C}_i$ be the set of network parameters in cluster $i$ and let $b_i$ be the number of bits of the codeword assigned to the network parameters in cluster $i$ for $1 \leq i \leq k$. For a lookup table to decode quantized

values from their binary encoded codewords, we store $k$ binary codewords ($b_i$ bits for $1 \le i \le k$) and corresponding quantized values ($b$ bits for each). The compression ratio is then given by

$$\text{Compression ratio} = \frac{Nb}{\sum_{i=1}^{k}(|\mathcal{C}_i| + 1)b_i + kb}. \tag{1}$$

Observe in (1) that the compression ratio depends not only on the number of clusters but also on the sizes of the clusters and the lengths of the binary codewords assigned to them, in particular, when a variable-length code is used for encoding quantized values. For fixed-length codes, however, all codewords are of the same length, i.e., $b_i = \lceil \log_2 k \rceil$ for all $1 \le i \le k$, and thus the compression ratio is reduced to only a function of the number of clusters, i.e., $k$, assuming that $N$ and $b$ are given.

## 2.2 K-MEANS CLUSTERING

Provided network parameters $\{w_i\}_{i=1}^{N}$ to quantize, k-means clustering partitions them into $k$ disjoint sets (clusters), denoted by $\mathcal{C}_1, \mathcal{C}_2, \ldots, \mathcal{C}_k$, while minimizing the mean square quantization error (MSQE) as follows:

$$\underset{\mathcal{C}_1,\mathcal{C}_2,\ldots,\mathcal{C}_k}{\text{argmin}} \sum_{i=1}^{k} \sum_{w \in \mathcal{C}_i} |w - c_i|^2, \quad \text{where} \quad c_i = \frac{1}{|\mathcal{C}_i|} \sum_{w \in \mathcal{C}_i} w. \tag{2}$$

We observe two issues with employing k-means clustering for network quantization.

- First, although k-means clustering minimizes the MSQE, it does not imply that k-means clustering minimizes the performance loss due to quantization as well in neural networks. K-means clustering treats quantization errors from all network parameters with equal importance. However, quantization errors from some network parameters may degrade the performance more significantly that the others. Thus, for minimizing the loss due to quantization in neural networks, one needs to take this dissimilarity into account.

- Second, k-means clustering does not consider any compression ratio constraint. It simply minimizes its distortion measure for a given number of clusters, i.e., for $k$ clusters. This is however suboptimal when variable-length coding follows since the compression ratio depends not only on the number of clusters but also on the sizes of the clusters and assigned codeword lengths to them, which are determined by the binary coding scheme employed after clustering. Therefore, for the optimization of network quantization given a compression ratio constraint, one need to take the impact of binary coding into account, i.e., we need to solve the quantization problem under the actual compression ratio constraint imposed by the specific binary coding scheme employed after clustering.

## 3 HESSIAN-WEIGHTED NETWORK QUANTIZATION

In this section, we analyze the impact of quantization errors on the neural network loss function and derive that the Hessian-weighted distortion measure is a relevant objective function for network quantization in order to minimize the quantization loss locally. Moreover, from this analysis, we propose Hessian-weighted k-means clustering for network quantization to minimize the performance loss due to quantization in neural networks.

## 3.1 NETWORK MODEL

We consider a general non-linear neural network that yields output $\mathbf{y} = f(\mathbf{x}; \mathbf{w})$ from input $\mathbf{x}$, where $\mathbf{w} = [w_1 \cdots w_N]^T$ is the vector consisting of all trainable network parameters in the network; $N$ is the total number of trainable parameters in the network. A loss function $loss(\mathbf{y}, \hat{\mathbf{y}})$ is defined as the objective function that we aim to minimize in average, where $\hat{\mathbf{y}} = \hat{\mathbf{y}}(\mathbf{x})$ is the expected (ground-truth) output for input $\mathbf{x}$. Cross entropy or mean square error are typical examples of a loss function. Given a training data set $\mathcal{X}_{\text{train}}$, we optimize network parameters by solving the following problem, e.g., approximately by using a stochastic gradient descent (SGD) method with mini-batches:

$$\hat{\mathbf{w}} = \underset{\mathbf{w}}{\text{argmin}}\, L(\mathcal{X}_{\text{train}}; \mathbf{w}), \quad \text{where} \quad L(\mathcal{X}; \mathbf{w}) = \frac{1}{|\mathcal{X}|} \sum_{\mathbf{x} \in \mathcal{X}} loss(f(\mathbf{x}; \mathbf{w}), \hat{\mathbf{y}}(\mathbf{x})).$$

## 3.2 HESSIAN-WEIGHTED QUANTIZATION ERROR

The average loss function $L(\mathcal{X}; \mathbf{w})$ can be expanded by Taylor series with respect to $\mathbf{w}$ as follows:

$$\delta L(\mathcal{X}; \mathbf{w}) = \mathbf{g}(\mathbf{w})^T \delta \mathbf{w} + \frac{1}{2} \delta \mathbf{w}^T \mathbf{H}(\mathbf{w}) \delta \mathbf{w} + O(\|\delta \mathbf{w}\|^3), \tag{3}$$

where

$$\mathbf{g}(\mathbf{w}) = \frac{\partial L(\mathcal{X}; \mathbf{w})}{\partial \mathbf{w}}, \quad \mathbf{H}(\mathbf{w}) = \frac{\partial^2 L(\mathcal{X}; \mathbf{w})}{\partial \mathbf{w}^2};$$

the square matrix $\mathbf{H}(\mathbf{w})$ consisting of second-order partial derivatives is called as Hessian matrix or Hessian. Assume that the loss function has reached to one of its local minima, at $\mathbf{w} = \hat{\mathbf{w}}$, after training. At local minima, gradients are all zero, i.e., we have $\mathbf{g}(\hat{\mathbf{w}}) = \mathbf{0}$, and thus the first term in the right-hand side of (3) can be neglected at $\mathbf{w} = \hat{\mathbf{w}}$. The third term in the right-hand side of (3) is also ignored under the assumption that the average loss function is approximately quadratic at the local minimum $\mathbf{w} = \hat{\mathbf{w}}$. Finally, for simplicity, we approximate the Hessian matrix as a diagonal matrix by setting its off-diagonal terms to be zero. Then, it follows from (3) that

$$\delta L(\mathcal{X}; \hat{\mathbf{w}}) \approx \frac{1}{2} \sum_{i=1}^{N} h_{ii}(\hat{\mathbf{w}}) |\delta \hat{w}_i|^2, \tag{4}$$

where $h_{ii}(\hat{\mathbf{w}})$ is the second-order partial derivative of the average loss function with respect to $w_i$ evaluated at $\mathbf{w} = \hat{\mathbf{w}}$, which is the $i$-th diagonal element of the Hessian matrix $\mathbf{H}(\hat{\mathbf{w}})$.

Now, we connect (4) with the problem of network quantization by treating $\delta \hat{w}_i$ as the quantization error of network parameter $w_i$ at its local optimum $w_i = \hat{w}_i$, i.e.,

$$\delta \hat{w}_i = \bar{w}_i - \hat{w}_i, \tag{5}$$

where $\bar{w}_i$ is a quantized value of $\hat{w}_i$. Finally, combining (4) and (5), we derive that the local impact of quantization on the average loss function at $\mathbf{w} = \hat{\mathbf{w}}$ can be quantified approximately as follows:

$$\delta L(\mathcal{X}; \hat{\mathbf{w}}) \approx \frac{1}{2} \sum_{i=1}^{N} h_{ii}(\hat{\mathbf{w}}) |\hat{w}_i - \bar{w}_i|^2. \tag{6}$$

At a local minimum, the diagonal elements of Hessian, i.e., $h_{ii}(\hat{\mathbf{w}})$'s, are all non-negative and thus the summation in (6) is always additive, implying that the average loss function either increases or stays the same. Therefore, the performance degradation due to quantization of a neural network can be measured approximately by the Hessian-weighted distortion as shown in (6). Further discussion on the Hessian-weighted distortion measure can be found in Appendix A.1.

## 3.3 HESSIAN-WEIGHTED K-MEANS CLUSTERING

For notational simplicity, we use $w_i \equiv \hat{w}_i$ and $h_{ii} \equiv h_{ii}(\hat{\mathbf{w}})$ from now on. The optimal clustering that minimizes the Hessian-weighted distortion measure is given by

$$\underset{\mathcal{C}_1, \mathcal{C}_2, \ldots, \mathcal{C}_k}{\operatorname{argmin}} \sum_{j=1}^{k} \sum_{w_i \in \mathcal{C}_j} h_{ii} |w_i - c_j|^2, \quad \text{where} \quad c_j = \frac{\sum_{w_i \in \mathcal{C}_j} h_{ii} w_i}{\sum_{w_i \in \mathcal{C}_j} h_{ii}}. \tag{7}$$

We call this as Hessian-weighted k-means clustering. Observe in (7) that we give a larger penalty for a network parameter in defining the distortion measure for clustering when its second-order partial derivative is larger, in order to avoid a large deviation from its original value, since the impact on the loss function due to quantization is expected to be larger for that parameter.

Hessian-weighted k-means clustering is locally optimal in minimizing the quantization loss when fixed-length binary coding follows, where the compression ratio solely depends on the number of clusters as shown in Section 2.1. Similar to the conventional k-means clustering, solving this optimization is not easy, but Lloyd's algorithm is still applicable as an efficient heuristic solution for this problem if Hessian-weighted means are used as cluster centers instead of non-weighted regular means.

## 3.4 HESSIAN COMPUTATION

For obtaining Hessian, one needs to evaluate the second-order partial derivative of the average loss function with respect to each of network parameters, i.e., we need to calculate

$$h_{ii}(\hat{\mathbf{w}}) = \left.\frac{\partial^2 L(\mathcal{X}; \mathbf{w})}{\partial w_i^2}\right|_{\mathbf{w}=\hat{\mathbf{w}}} = \frac{1}{|\mathcal{X}|}\frac{\partial^2}{\partial w_i^2}\sum_{\mathbf{x}\in\mathcal{X}} loss(f(\mathbf{x}; \mathbf{w}), \hat{\mathbf{y}}(\mathbf{x}))\bigg|_{\mathbf{w}=\hat{\mathbf{w}}}. \tag{8}$$

Recall that we are interested in only the diagonal elements of Hessian. An efficient way of computing the diagonal of Hessian is presented in Le Cun (1987); Becker & Le Cun (1988) and it is based on the back propagation method that is similar to the back propagation algorithm used for computing first-order partial derivatives (gradients). That is, computing the diagonal of Hessian is of the same order of complexity as computing gradients.

Hessian computation and our network quantization are performed after completing network training. For the data set $\mathcal{X}$ used to compute Hessian in (8), we can either reuse a training data set or use some other data set, e.g., validation data set. We observed from our experiments that even using a small subset of the training or validation data set is sufficient to yield good approximation of Hessian for network quantization.

## 3.5 ALTERNATIVE OF HESSIAN

Although there is an efficient way to obtain the diagonal of Hessian as discussed in the previous subsection, Hessian computation is not free. In order to avoid this additional Hessian computation, we propose to use an alternative metric instead of Hessian. In particular, we consider neural networks trained with the Adam SGD optimizer (Kingma & Ba, 2014) and propose to use some function (e.g., square root) of the second moment estimates of gradients as an alternative of Hessian.

The Adam algorithm computes adaptive learning rates for individual network parameters from the first and second moment estimates of gradients. We compare the Adam method to Newton's optimization method using Hessian and notice that the second moment estimates of gradients in the Adam method act like the Hessian in Newton's method. This observation leads us to use some function (e.g., square root) of the second moment estimates of gradients as an alternative of Hessian.

The advantage of using the second moment estimates from the Adam method is that they are computed while training and we can obtain them at the end of training at no additional cost. It makes Hessian-weighting more feasible for deep neural networks, which have millions of parameters. We note that similar quantities can be found and used for other SGD optimization methods using adaptive learning rates, e.g., AdaGrad (Duchi et al., 2011), Adadelta (Zeiler, 2012) and RMSProp (Tieleman & Hinton, 2012).

## 3.6 QUANTIZATION OF ALL LAYERS

We propose quantizing the network parameters of all layers in a neural network together at once by taking Hessian-weight into account. Layer-by-layer quantization was examined in the previous work (Gong et al., 2014; Han et al., 2015a). However, e.g., in Han et al. (2015a), a larger number of bits (a larger number of clusters) are assigned to convolutional layers than fully-connected layers, which implies that they heuristically treat convolutional layers more importantly. This follows from the fact that the impact of quantization errors on the performance varies significantly across layers; some layers, e.g., convolutional layers, may be more important than the others. This concern is exactly what we can address by Hessian-weighting.

Hessian-weighting properly handles the different impact of quantization errors not only within layers but also across layers and thus it can be employed for quantizing all layers of a network together. The impact of quantization errors may vary more substantially across layers than within layers. Thus, Hessian-weighting may show more benefit in deeper neural networks. We note that Hessian-weighting can still provide gain even for layer-by-layer quantization since it can address the different impact of the quantization errors of network parameters within each layer as well.

Recent neural networks are getting deeper, e.g., see Szegedy et al. (2015a;b); He et al. (2015). For such deep neural networks, quantizing network parameters of all layers together is even more advantageous since we can avoid layer-by-layer compression rate optimization. Optimizing compression

ratios jointly across all individual layers (to maximize the overall compression ratio for a network) requires exponential time complexity with respect to the number of layers. This is because the total number of possible combinations of compression ratios for individual layers increases exponentially as the number of layers increases.

## 4 ENTROPY-CONSTRAINED NETWORK QUANTIZATION

In this section, we investigate how to solve the network quantization problem under a constraint on the compression ratio. In designing network quantization schemes, we not only want to minimize the performance loss but also want to maximize the compression ratio. In Section 3, we explored how to quantify and minimize the loss due to quantization. In this section, we investigate how to take the compression ratio into account properly in the optimization of network quantization.

### 4.1 ENTROPY CODING

After quantizing network parameters by clustering, lossless data compression by variable-length binary coding can be followed for compressing quantized values. There is a set of optimal codes that achieve the minimum average codeword length for a given source. Entropy is the theoretical limit of the average codeword length per symbol that we can achieve by lossless data compression, proved by Shannon (see, e.g., Cover & Thomas (2012, Section 5.3)). It is known that optimal codes achieve this limit with some overhead less than 1 bit when only integer-length codewords are allowed. So optimal coding is also called as entropy coding. Huffman coding is one of entropy coding schemes commonly used when the source distribution is provided (see, e.g., Cover & Thomas (2012, Section 5.6)), or can be estimated.

### 4.2 ENTROPY-CONSTRAINED SCALAR QUANTIZATION (ECSQ)

Considering a compression ratio constraint in network quantization, we need to solve the clustering problem in (2) or (7) under the compression ratio constraint given by

$$\text{Compression ratio} = \frac{b}{\bar{b} + (\sum_{i=1}^{k} b_i + kb)/N} > C, \quad \text{where} \quad \bar{b} = \frac{1}{N} \sum_{i=1}^{k} |\mathcal{C}_i| b_i, \tag{9}$$

which follows from (1). This optimization problem is too complex to solve for any arbitrary variable-length binary code since the average codeword length $\bar{b}$ can be arbitrary. However, we identify that it can be simplified if optimal codes, e.g., Huffman codes, are assumed to be used. In particular, optimal coding closely achieves the lower limit of the average source code length, i.e., entropy, and then we approximately have

$$\bar{b} \approx H = - \sum_{i=1}^{k} p_i \log_2 p_i, \tag{10}$$

where $H$ is the entropy of the quantized network parameters after clustering (i.e., source), given that $p_i = |\mathcal{C}_i|/N$ is the ratio of the number of network parameters in cluster $\mathcal{C}_i$ to the number of all network parameters (i.e., source distribution). Moreover, assuming that $N \gg k$, we have

$$\frac{1}{N} \left( \sum_{i=1}^{k} b_i + kb \right) \approx 0, \tag{11}$$

in (9). From (10) and (11), the constraint in (9) can be altered to an entropy constraint given by

$$H = - \sum_{i=1}^{k} p_i \log_2 p_i < R,$$

where $R \approx b/C$. In summary, assuming that optimal coding is employed after clustering, one can approximately replace a compression ratio constraint with an entropy constraint for the clustering output. The network quantization problem is then translated into a quantization problem with an entropy constraint, which is called as entropy-constrained scalar quantization (ECSQ) in information theory. Two efficient heuristic solutions for ECSQ are proposed for network quantization in the following subsections, i.e., uniform quantization and an iterative solution similar to Lloyd's algorithm for k-means clustering.

### 4.3 Uniform quantization

It is shown in Gish & Pierce (1968) that the uniform quantizer is asymptotically optimal in minimizing the mean square quantization error for any random source with a reasonably smooth density function as the resolution becomes infinite, i.e., as the number of clusters $k \to \infty$. This asymptotic result leads us to come up with a very simple but efficient network quantization scheme as follows:

1. We first set uniformly spaced thresholds and divide network parameters into clusters.
2. After determining clusters, their quantized values (cluster centers) are obtained by taking the mean of network parameters in each cluster.

Note that one can use Hessian-weighted mean instead of non-weighted mean in computing cluster centers in the second step above in order to take the benefit of Hessian-weighting. A performance comparison of uniform quantization with non-weighted mean and uniform quantization with Hessian-weighted mean can be found in Appendix A.2.

Although uniform quantization is a straightforward method, it has never been shown before in the literature that it is actually one of the most efficient quantization schemes for neural networks when optimal variable-length coding, e.g., Huffman coding, follows. We note that uniform quantization is not always good; it is inefficient for fixed-length coding, which is also first shown in this paper.

### 4.4 Iterative algorithm to solve ECSQ

Another scheme proposed to solve the ECSQ problem for network quantization is an iterative algorithm, which is similar to Lloyd's algorithm for k-means clustering. Although this iterative solution is more complicated than the uniform quantization in Section 4.3, it finds a local optimum for a given discrete source. An iterative algorithm to solve the general ECSQ problem is provided in Chou et al. (1989). We derive a similar iterative algorithm to solve the ECSQ problem for network quantization. The main difference from the method in Chou et al. (1989) is that we minimize the Hessian-weighted distortion measure instead of the non-weighted regular distortion measure for optimal quantization. The detailed algorithm and further discussion can be found in Appendix A.3.

## 5 Experiments

This section presents our experiment results for the proposed network quantization schemes in three exemplary convolutional neural networks: (a) LeNet (LeCun et al., 1998) for the MNIST data set, (b) ResNet (He et al., 2015) for the CIFAR-10 data set, and (c) AlexNet (Krizhevsky et al., 2012) for the ImageNet ILSVRC-2012 data set. Our experiments can be summarized as follows:

- We employ the proposed network quantization methods to quantize all of network parameters in a network together at once, as discussed in Section 3.6.
- We evaluate the performance of the proposed network quantization methods with and without network pruning. For a pruned model, we need to store not only the values of unpruned parameters but also their respective indexes (locations) in the original model. For the index information, we compute index differences between unpruned network parameters in the original model and further compress them by Huffman coding as in Han et al. (2015a).
- For Hessian computation, 50,000 samples of the training set are reused. We also evaluate the performance when Hessian is computed with 1,000 samples only.
- Finally, we evaluate the performance of our network quantization schemes using Hessian when its alternative is used instead, as discussed in Section 3.5. To this end, we retrain the considered neural networks with the Adam SGD optimizer and obtain the second moment estimates of gradients at the end of training. Then, we use the square roots of the second moment estimates instead of Hessian and evaluate the performance.

### 5.1 Experiment models

First, we evaluate our network quantization schemes for the MNIST data set with a simplified version of LeNet5 (LeCun et al., 1998), consisting of two convolutional layers and two fully-connected

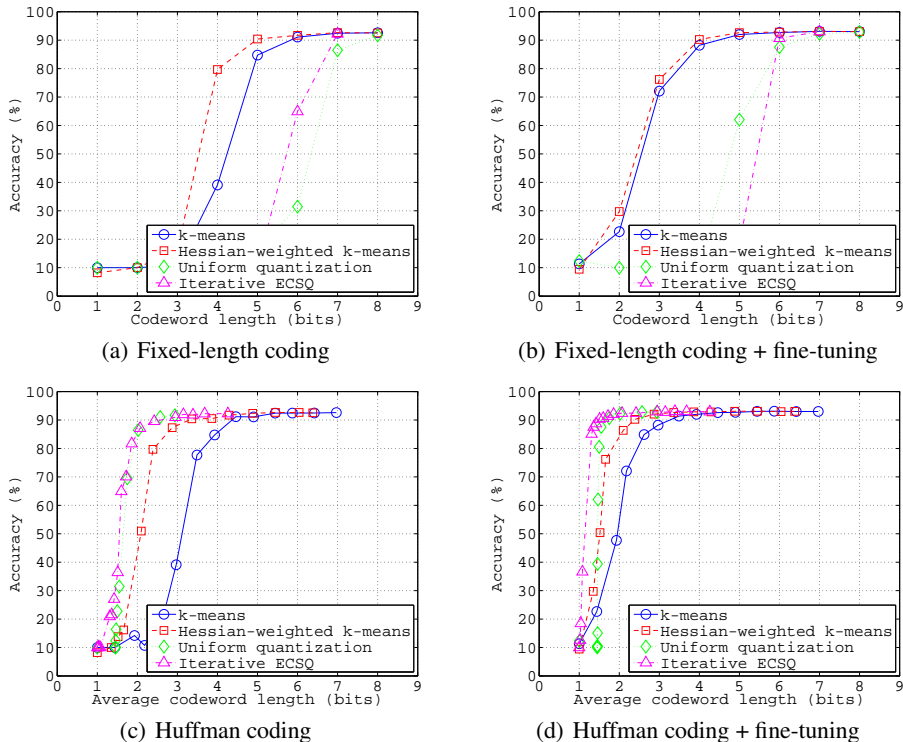

Figure 1: Accuracy versus average codeword length per network parameter after network quantization for 32-layer ResNet.

layers followed by a soft-max layer. It has total 431,080 parameters and achieves 99.25% accuracy. For a pruned model, we prune 91% of the original network parameters and fine-tune the rest.

Second, we experiment our network quantization schemes for the CIFAR-10 data set (Krizhevsky, 2009) with a pre-trained 32-layer ResNet (He et al., 2015). The 32-layer ResNet consists of 464,154 parameters in total and achieves 92.58% accuracy. For a pruned model, we prune 80% of the original network parameters and fine-tune the rest.

Third, we evaluate our network quantization schemes with AlexNet (Krizhevsky et al., 2012) for the ImageNet ILSVRC-2012 data set (Russakovsky et al., 2015). We obtain a pre-trained AlexNet Caffe model, which achieves 57.16% top-1 accuracy. For a pruned model, we prune 89% parameters and fine-tune the rest. In fine-tuning, the Adam SGD optimizer is used in order to avoid the computation of Hessian by utilizing its alternative (see Section 3.5). However, the pruned model does not recover the original accuracy after fine-tuning with the Adam method; the top-1 accuracy recovered after pruning and fine-tuning is 56.00%. We are able to find a better pruned model achieving the original accuracy by pruning and retraining iteratively (Han et al., 2015b), which is however not used here.

## 5.2 EXPERIMENT RESULTS

We first present the quantization results without pruning for 32-layer ResNet in Figure 1, where the accuracy of 32-layer ResNet is plotted against the average codeword length per network parameter after quantization. When fixed-length coding is employed, the proposed Hessian-weighted k-means clustering method performs the best, as expected. Observe that Hessian-weighted k-means clustering yields better accuracy than others even after fine-tuning. On the other hand, when Huffman coding is employed, uniform quantization and the iterative algorithm for ECSQ outperform Hessian-weighted k-means clustering and k-means clustering. However, these two ECSQ solutions underperform Hessian-weighted k-means clustering and even k-means clustering when fixed-length coding is employed since they are optimized for optimal variable-length coding.

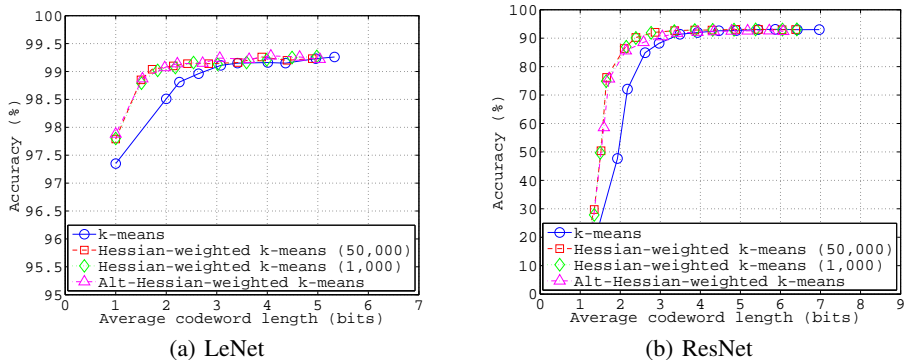

(a) LeNet  (b) ResNet

Figure 2: Accuracy versus average codeword length per network parameter after network quantization, Huffman coding and fine-tuning for LeNet and 32-layer ResNet when Hessian is computed with 50,000 or 1,000 samples and when the square roots of the second moment estimates of gradients are used instead of Hessian as an alternative.

Figure 2 shows the performance of Hessian-weighted k-means clustering when Hessian is computed with a small number of samples (1,000 samples). Observe that even using the Hessian computed with a small number of samples yields almost the same performance. We also show the performance of Hessian-weighted k-means clustering when an alternative of Hessian is used instead of Hessian as explained in Section 3.5. In particular, the square roots of the second moment estimates of gradients are used instead of Hessian, and using this alternative provides similar performance to using Hessian.

In Table 1, we summarize the compression ratios that we can achieve with different network quantization methods for pruned models. The original network parameters are 32-bit float numbers. Using the simple uniform quantization followed by Huffman coding, we achieve the compression ratios of 51.25, 22.17 and 40.65 (i.e., the compressed model sizes are 1.95%, 4.51% and 2.46% of the original model sizes) for LeNet, 32-layer ResNet and AlexNet, respectively, at no or marginal performance loss. Observe that the loss in the compressed AlexNet is mainly due to pruning. Here, we also compare our network quantization results to the ones in Han et al. (2015a). Note that layer-by-layer quantization with k-means clustering is evaluated in Han et al. (2015a) while our quantization schemes including k-means clustering are employed to quantize network parameters of all layers together at once (see Section 3.6).

## 6 CONCLUSION

This paper investigates the quantization problem of network parameters in deep neural networks. We identify the suboptimality of the conventional quantization method using k-means clustering and newly design network quantization schemes so that they can minimize the performance loss due to quantization given a compression ratio constraint. In particular, we analytically show that Hessian can be used as a measure of the importance of network parameters and propose to minimize Hessian-weighted quantization errors in average for clustering network parameters to quantize. Hessian-weighting is beneficial in quantizing all of the network parameters together at once since it can handle the different impact of quantization errors properly not only within layers but also across layers. Furthermore, we make a connection from the network quantization problem to the entropy-constrained data compression problem in information theory and push the compression ratio to the limit that information theory provides. Two efficient heuristic solutions are presented to this end, i.e., uniform quantization and an iterative solution for ECSQ. Our experiment results show that the proposed network quantization schemes provide considerable gain over the conventional method using k-means clustering, in particular for large and deep neural networks.

## REFERENCES

Sajid Anwar, Kyuyeon Hwang, and Wonyong Sung. Fixed point optimization of deep convolutional neural networks for object recognition. In *IEEE International Conference on Acoustics, Speech*

Table 1: Summary of network quantization results with Huffman coding for pruned models.

| | | | Accuracy % | Compression ratio |
|---|---|---|---|---|
| LeNet | Original model | | 99.25 | - |
| | Pruned model | | 99.27 | 10.13 |
| | Pruning + Quantization all layers + Huffman coding | k-means | 99.27 | 44.58 |
| | | Hessian-weighted k-means | 99.27 | 47.16 |
| | | Uniform quantization | 99.28 | 51.25 |
| | | Iterative ECSQ | 99.27 | 49.01 |
| | Deep compression (Han et al., 2015a) | | 99.26 | 39.00 |
| ResNet | Original model | | 92.58 | - |
| | Pruned model | | 92.58 | 4.52 |
| | Pruning + Quantization all layers + Huffman coding | k-means | 92.64 | 18.25 |
| | | Hessian-weighted k-means | 92.67 | 20.51 |
| | | Uniform quantization | 92.68 | 22.17 |
| | | Iterative ECSQ | 92.73 | 21.01 |
| | Deep compression (Han et al., 2015a) | | N/A | N/A |
| AlexNet | Original model | | 57.16 | - |
| | Pruned model | | 56.00 | 7.91 |
| | Pruning + Quantization all layers + Huffman coding | k-means | 56.12 | 30.53 |
| | | Alt-Hessian-weighted k-means | 56.04 | 33.71 |
| | | Uniform quantization | 56.20 | 40.65 |
| | Deep compression (Han et al., 2015a) | | 57.22 | 35.00 |

*and Signal Processing*, pp. 1131–1135, 2015.

Sue Becker and Yann Le Cun. Improving the convergence of back-propagation learning with second order methods. In *Proceedings of the Connectionist Models Summer School*, pp. 29–37. San Matteo, CA: Morgan Kaufmann, 1988.

Philip A Chou, Tom Lookabaugh, and Robert M Gray. Entropy-constrained vector quantization. *IEEE Transactions on Acoustics, Speech, and Signal Processing*, 37(1):31–42, 1989.

Matthieu Courbariaux, Jean-Pierre David, and Yoshua Bengio. Training deep neural networks with low precision multiplications. *arXiv preprint arXiv:1412.7024*, 2014.

Matthieu Courbariaux, Yoshua Bengio, and Jean-Pierre David. Binaryconnect: Training deep neural networks with binary weights during propagations. In *Advances in Neural Information Processing Systems*, pp. 3123–3131, 2015.

Thomas M Cover and Joy A Thomas. *Elements of information theory*. John Wiley & Sons, 2012.

John Duchi, Elad Hazan, and Yoram Singer. Adaptive subgradient methods for online learning and stochastic optimization. *Journal of Machine Learning Research*, 12(Jul):2121–2159, 2011.

Herbert Gish and John Pierce. Asymptotically efficient quantizing. *IEEE Transactions on Information Theory*, 14(5):676–683, 1968.

Yunchao Gong, Liu Liu, Ming Yang, and Lubomir Bourdev. Compressing deep convolutional networks using vector quantization. *arXiv preprint arXiv:1412.6115*, 2014.

Suyog Gupta, Ankur Agrawal, Kailash Gopalakrishnan, and Pritish Narayanan. Deep learning with limited numerical precision. In *Proceedings of the 32nd International Conference on Machine Learning*, pp. 1737–1746, 2015.

Song Han, Huizi Mao, and William J Dally. Deep compression: Compressing deep neural networks with pruning, trained quantization and huffman coding. *arXiv preprint arXiv:1510.00149*, 2015a.

Song Han, Jeff Pool, John Tran, and William Dally. Learning both weights and connections for efficient neural network. In *Advances in Neural Information Processing Systems*, pp. 1135–1143, 2015b.

Babak Hassibi and David G Stork. Second order derivatives for network pruning: Optimal brain surgeon. In *Advances in Neural Information Processing Systems*, pp. 164–171, 1993.

Kaiming He, Xiangyu Zhang, Shaoqing Ren, and Jian Sun. Deep residual learning for image recognition. *arXiv preprint arXiv:1512.03385*, 2015.

Max Jaderberg, Andrea Vedaldi, and Andrew Zisserman. Speeding up convolutional neural networks with low rank expansions. In *Proceedings of the British Machine Vision Conference*, 2014.

Yong-Deok Kim, Eunhyeok Park, Sungjoo Yoo, Taelim Choi, Lu Yang, and Dongjun Shin. Compression of deep convolutional neural networks for fast and low power mobile applications. *arXiv preprint arXiv:1511.06530*, 2015.

Diederik Kingma and Jimmy Ba. Adam: A method for stochastic optimization. *arXiv preprint arXiv:1412.6980*, 2014.

Alex Krizhevsky. Learning multiple layers of features from tiny images. 2009.

Alex Krizhevsky, Ilya Sutskever, and Geoffrey E Hinton. Imagenet classification with deep convolutional neural networks. In *Advances in Neural Information Processing Systems*, pp. 1097–1105, 2012.

Yann Le Cun. *Modèles connexionnistes de l'apprentissage*. PhD thesis, Paris 6, 1987.

Vadim Lebedev and Victor Lempitsky. Fast convnets using group-wise brain damage. In *Proceedings of the IEEE Conference on Computer Vision and Pattern Recognition*, pp. 2554–2564, 2016.

Vadim Lebedev, Yaroslav Ganin, Maksim Rakhuba, Ivan Oseledets, and Victor Lempitsky. Speeding-up convolutional neural networks using fine-tuned CP-decomposition. *arXiv preprint arXiv:1412.6553*, 2014.

Yann LeCun, John S Denker, Sara A Solla, Richard E Howard, and Lawrence D Jackel. Optimal brain damage. In *Advances in Neural Information Processing Systems*, pp. 598–605, 1989.

Yann LeCun, Léon Bottou, Yoshua Bengio, and Patrick Haffner. Gradient-based learning applied to document recognition. *Proceedings of the IEEE*, 86(11):2278–2324, 1998.

Yann LeCun, Yoshua Bengio, and Geoffrey Hinton. Deep learning. *Nature*, 521(7553):436–444, 2015.

Darryl D Lin, Sachin S Talathi, and V Sreekanth Annapureddy. Fixed point quantization of deep convolutional networks. *arXiv preprint arXiv:1511.06393*, 2015a.

Zhouhan Lin, Matthieu Courbariaux, Roland Memisevic, and Yoshua Bengio. Neural networks with few multiplications. *arXiv preprint arXiv:1510.03009*, 2015b.

Baoyuan Liu, Min Wang, Hassan Foroosh, Marshall Tappen, and Marianna Pensky. Sparse convolutional neural networks. In *Proceedings of the IEEE Conference on Computer Vision and Pattern Recognition*, pp. 806–814, 2015.

Michael C Mozer and Paul Smolensky. Skeletonization: A technique for trimming the fat from a network via relevance assessment. In *Advances in Neural Information Processing Systems*, pp. 107–115, 1989.

Alexander Novikov, Dmitrii Podoprikhin, Anton Osokin, and Dmitry P Vetrov. Tensorizing neural networks. In *Advances in Neural Information Processing Systems*, pp. 442–450, 2015.

Mohammad Rastegari, Vicente Ordonez, Joseph Redmon, and Ali Farhadi. XNOR-Net: Imagenet classification using binary convolutional neural networks. *arXiv preprint arXiv:1603.05279*, 2016.

Olga Russakovsky, Jia Deng, Hao Su, Jonathan Krause, Sanjeev Satheesh, Sean Ma, Zhiheng Huang, Andrej Karpathy, Aditya Khosla, Michael Bernstein, et al. Imagenet large scale visual recognition challenge. *International Journal of Computer Vision*, 115(3):211–252, 2015.

Tara N Sainath, Brian Kingsbury, Vikas Sindhwani, Ebru Arisoy, and Bhuvana Ramabhadran. Low-rank matrix factorization for deep neural network training with high-dimensional output targets. In *IEEE International Conference on Acoustics, Speech and Signal Processing*, pp. 6655–6659, 2013.

Karen Simonyan and Andrew Zisserman. Very deep convolutional networks for large-scale image recognition. *arXiv preprint arXiv:1409.1556*, 2014.

Christian Szegedy, Wei Liu, Yangqing Jia, Pierre Sermanet, Scott Reed, Dragomir Anguelov, Dumitru Erhan, Vincent Vanhoucke, and Andrew Rabinovich. Going deeper with convolutions. In *Proceedings of the IEEE Conference on Computer Vision and Pattern Recognition*, pp. 1–9, 2015a.

Christian Szegedy, Vincent Vanhoucke, Sergey Ioffe, Jonathon Shlens, and Zbigniew Wojna. Rethinking the inception architecture for computer vision. *arXiv preprint arXiv:1512.00567*, 2015b.

Cheng Tai, Tong Xiao, Xiaogang Wang, et al. Convolutional neural networks with low-rank regularization. *arXiv preprint arXiv:1511.06067*, 2015.

Tijmen Tieleman and Geoffrey Hinton. Lecture 6.5-rmsprop: Divide the gradient by a running average of its recent magnitude. *COURSERA: Neural Networks for Machine Learning*, 4(2), 2012.

Vincent Vanhoucke, Andrew Senior, and Mark Z Mao. Improving the speed of neural networks on CPUs. In *Deep Learning and Unsupervised Feature Learning Workshop, NIPS*, 2011.

Wei Wen, Chunpeng Wu, Yandan Wang, Yiran Chen, and Hai Li. Learning structured sparsity in deep neural networks. In *Advances in Neural Information Processing Systems*, pp. 2074–2082, 2016.

Jian Xue, Jinyu Li, and Yifan Gong. Restructuring of deep neural network acoustic models with singular value decomposition. In *INTERSPEECH*, pp. 2365–2369, 2013.

Zichao Yang, Marcin Moczulski, Misha Denil, Nando de Freitas, Alex Smola, Le Song, and Ziyu Wang. Deep fried convnets. In *Proceedings of the IEEE International Conference on Computer Vision*, pp. 1476–1483, 2015.

Matthew D Zeiler. Adadelta: an adaptive learning rate method. *arXiv preprint arXiv:1212.5701*, 2012.

# A   APPENDIX

## A.1   FURTHER DISCUSSION ON THE HESSIAN-WEIGHTED QUANTIZATION ERROR

The diagonal approximation for Hessian simplifies the optimization problem as well as its solution for network quantization. This simplification comes with some performance loss. We conjecture that the loss due to this approximation is small. The reason is that the contributions from off-diagonal terms are not always additive and their summation may end up with a small value. However, diagonal terms are all non-negative and therefore their contributions are always additive. We do not verify this conjecture in this paper since solving the problem without diagonal approximation is too complex; we even need to compute the whole Hessian matrix, which is also too costly.

Observe that the relation of the Hessian-weighted distortion measure to the quantization loss holds for any model for which the objective function can be approximated as a quadratic function with respect to the parameters to quantize in the model. Hence, the quantization methods proposed in this paper to minimize the Hessian-weighted distortion measure are not specific to neural networks but are generally applicable to quantization of parameters of any model whose objective function is locally quadratic with respect to its parameters approximately.

Finally, we do not consider the interactions between quantization and retraining in our formulation in Section 3.2. We analyze the expected loss due to quantization assuming no further retraining and focus on finding optimal network quantization schemes that minimize the performance loss. In our experiments, however, we further fine-tune the quantized values (cluster centers) so that we can recover the loss due to quantization and improve the performance.

## A.2   EXPERIMENT RESULTS FOR UNIFORM QUANTIZATION

We compare uniform quantization with non-weighted mean and uniform quantization with Hessian-weighted mean in Figure 3, which shows that uniform quantization with Hessian-weighted mean slightly outperforms uniform quantization with non-weighted mean.

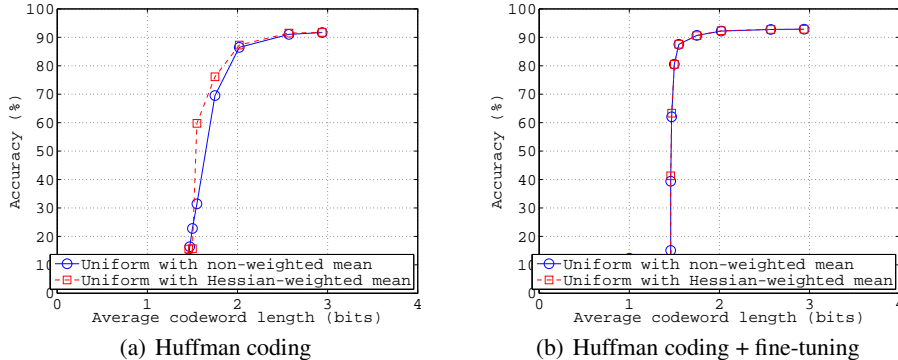

(a) Huffman coding (b) Huffman coding + fine-tuning

Figure 3: Accuracy versus average codeword length per network parameter after network quantization, Huffman coding and fine-tuning for 32-layer ResNet when uniform quantization with non-weighted mean and uniform quantization with Hessian-weighted mean are used.

## A.3   FURTHER DISCUSSION ON THE ITERATIVE ALGORITHM FOR ECSQ

In order to solve the ECSQ problem for network quantization, we define a Lagrangian cost function:

$$J_\lambda(\mathcal{C}_1, \mathcal{C}_2, \ldots, \mathcal{C}_k) = D + \lambda H = \frac{1}{N} \sum_{j=1}^{k} \sum_{w_i \in \mathcal{C}_j} \underbrace{(h_{ii}|w_i - c_j|^2 - \lambda \log_2 p_j)}_{=d_\lambda(i,j)}, \quad (12)$$

where

$$D = \frac{1}{N} \sum_{j=1}^{k} \sum_{w_i \in \mathcal{C}_j} h_{ii}|w_i - c_j|^2, \quad H = -\sum_{j=1}^{k} p_j \log_2 p_j.$$

---

**Algorithm 1** Iterative solution for entropy-constrained network quantization

---

**Initialization:** $n \leftarrow 0$

Initialize the centers of $k$ clusters: $c_1^{(0)}, \ldots, c_k^{(0)}$

Initialize the proportions of $k$ clusters (set all of them to be the same initially): $p_1^{(0)}, \ldots, p_k^{(0)}$

**repeat**

 **Assignment:**

 **for all** network parameters $i = 1 \rightarrow N$ **do**

 Assign $w_i$ to the cluster $j$ that minimizes the individual Lagrangian cost as follows:

$$\mathcal{C}_l^{(n+1)} \leftarrow \mathcal{C}_l^{(n+1)} \cup \{w_i\} \quad \text{for} \quad l = \underset{j}{\operatorname{argmin}} \left\{ h_{ii}|w_i - c_j^{(n)}|^2 - \lambda \log_2 p_j^{(n)} \right\}$$

 **end for**

 **Update:**

 **for all** clusters $j = 1 \rightarrow k$ **do**

 Update the cluster center and the proportion of cluster $j$:

$$c_j^{(n+1)} \leftarrow \frac{\sum_{w_i \in \mathcal{C}_j^{(n+1)}} h_{ii} w_i}{\sum_{w_i \in \mathcal{C}_j^{(n+1)}} h_{ii}} \quad \text{and} \quad p_j^{(n+1)} \leftarrow \frac{|\mathcal{C}_j^{(n+1)}|}{N}$$

 **end for**

 $n \leftarrow n + 1$

**until** Lagrangian cost function $J_\lambda$ decreases less than some threshold

---

The entropy-constrained network quantization problem is then reduced to find $k$ partitions (clusters) $\mathcal{C}_1, \mathcal{C}_2, \ldots, \mathcal{C}_k$ that minimize the Lagrangian cost function as follows:

$$\underset{\mathcal{C}_1, \mathcal{C}_2, \ldots, \mathcal{C}_k}{\operatorname{argmin}} \quad J_\lambda(\mathcal{C}_1, \mathcal{C}_2, \ldots, \mathcal{C}_k).$$

A heuristic iterative algorithm to solve this method of Lagrange multipliers for network quantization is presented in Algorithm 1. It is similar to Lloyd's algorithm for k-means clustering. The key difference is how to partition network parameters at the assignment step. In Lloyd's algorithm, the Euclidean distance (quantization error) is minimized. For ECSQ, the individual Lagrangian cost function, i.e., $d_\lambda(i, j)$ in (12), is minimized instead, which includes both quantization error and expected codeword length after entropy coding.

