# Peer review of "Towards the Limit of Network Quantization"

_ICLR 2017 — accepted_

[Author Response · Yoojin Choi · 05 Dec 2016]
**Revision Dec-4**

Dear Reviewers,

We have upload a revised paper. Here is a list of summary for our revision.

1) We added a performance comparison of uniform quantization with non-weighed mean and uniform quantization with Hessian-weighted mean in Appendix A.3 in the revised paper. We also added a sentence for this in lines 2-4 of the last paragraph of Section 5.3 in the revised paper.

2) We revised the second paragraph of Section 4.5 in the original paper and clarified it as follows:

"We note that quantization of all layers is proposed under the assumption that all of binary encoded quantized parameters in a network are simply stored in one single array. Under this assumption, if layer-by-layer quantization is employed, then we need to assign some (additional) bits to each binary codeword for layer information (layer index), and it hurts the compression ratio. If we quantize all parameters of a network together, then we can avoid such additional overhead for layer indication when storing binary encoded quantized parameters. Thus, in this case, quantizing all layers together is beneficial and Hessian-weighting can be used to address the different impact of the quantization errors across layers.

For layer-by-layer quantization, it is advantageous to use separate arrays and separate lookup tables for different layers since layer information can be excluded in each of binary codewords for network parameters. Hessian-weighting can still provide gain even in this case for layer-by-layer quantization since it can address the different impact of the quantization errors of network parameters within each layer as well."

3) We added the following sentence in the abstract as your suggestion (see Abstract lines 12-15 in the revised paper).

"Hessian-weighting properly handles the different impact of quantization errors not only within layers but also across layers and thus it can be employed for quantizing all layers of a network together at once; it is beneficial since one can avoid layer-by-layer compression rate optimization."

We also added the following sentence in Section 7 as your suggestion (see Section 7 lines 8-11 in the revised paper).

"Hessian-weighting is beneficial in quantizing all of the network parameters together at once since it can handle the different impact of quantization errors properly not only within layers but also across layers; thus, using Hessian-weighting, we can avoid layer-by-layer compression rate optimization."

4) We reduced the number of pages from 17 to 15 (excluding reference and appendices).

- We removed the third paragraph in page 2 of the original paper.
- We removed the last two sentences in the first paragraph of Section 2 of the original paper.
- We removed the last three sentences in the first paragraph of Section 3 of the original paper.
- We removed the last three sentences in the first paragraph of Section 4 of the original paper.
- We removed the second to the fifth paragraphs in Section 5 of the original paper.
- We revised the paragraphs right before Section 6.1 and Section 6.1 as well in order to remove any duplications.
- We moved the second table in Table 1 of the original paper to Appendix A.4 in the revised paper since it is extra information.

5) We added the following sentence at the end of Section 4.1 of the revised paper for clarification.

"We note that we do not consider the interactions between retraining and quantization in our formulation. In this paper, we analyze the expected loss due to quantization of all network parameters assuming no further retraining and focus on finding optimal network quantization schemes that minimize the performance loss while maximizing the compression ratio. After quantization, however, in our experiments, we fine-tune the quantized values (cluster centers) so that we can recover the loss due to quantization and improve the performance further."

Thank you again for your comments and suggestions. We look forward to further comments.

Best,
Yoojin

[Official Review · AnonReviewer2 · rating 7 · confidence 3 · 16 Dec 2016]
**interesting experimental evaluation of variable bit-rate CNN weight compression scheme**

This paper proposes a novel neural network compression technique.
The goal is to compress maximally the network specification via parameter quantisation with a minimum impact on the expected loss.
It assumes pruning of the network parameters has already been performed, and only considers the quantisation of the individual scalar parameters of the network.
In contrast to previous work (Han et al. 2015a, Gong et al. 2014) the proposed approach takes into account the effect of the weight quantisation on the loss function that is used to train the network, and also takes into account the effect on a variable-length binary encoding of the cluster centers used for the quantisation. 

Unfortunately, the submitted paper is 20 pages, rather than the 8 recommended. The length of the paper seems unjustified to me, since the first three sections (first five pages) are very generic and redundant can be largely compressed or skipped (including figures 1 and 2). Although not a strict requirement by the submission guidelines, I would suggest the authors to compress their paper to 8 pages, this will improve the readability of the paper.

To take into account the impact on the network’s loss the authors propose to use a second order approximation of the cost function of the loss. In the case of weights that originally constitute a local minimum of the loss, this leads to a formulation of the impact of the weight quantization on the loss in terms of a weighted k-means clustering objective, where the weights are derived from the hessian of the loss function at the original weights.
The hessian can be computed efficiently using a back-propagation algorithm similar to that used to compute the gradient, as shown in cited work from the literature. 
The authors also propose to alternatively use a second-order moment term used by the Adam optimisation algorithm, since it can be loosely interpreted as an approximate Hessian. 

In section 4.5 the authors argue that with their approach it is more natural to quantise weights across all layers together, due to the hessian weighting which takes into account the variable impact across layers of quantisation errors on the network performance. 
The last statement in this section, however, was not clear to me: 
“In such deep neural networks, quantising network parameters of all layers together is more efficient since optimizing layer-by-layer clustering jointly across all layers requires exponential time complexity with respect to the number of layers.”
Perhaps the authors could elaborate a bit more on this point?

In section 5 the authors develop methods to take into account the code length of the weight quantisation in the clustering process. 
The first method described by the authors (based on previous work), is uniform quantisation of the weight space, which is then further optimised by their hessian-weighted clustering procedure from section 4. 
For the case of nonuniform codeword lengths to encode the cluster indices, the authors develop a modification of the Hessian weighted k-means algorithm in which the code length of each cluster is also taken into account, weighted by a factor lambda. Different values of lambda give rise to different compression-accuracy trade-offs, and the authors propose to cluster weights for a variety of lambda values and then pick the most accurate solution obtained, given a certain compression budget.  

In section 6 the authors report a number of experimental results that were obtained with the proposed methods, and compare these results to those obtained by the layer-wise compression technique of Han et al 2015, and to the uncompressed models. 
For these experiments the authors used three datasets, MNIST, CIFAR10 and ImageNet, with data-set specific architectures taken from the literature. 
These results suggest a consistent and significant advantage of the proposed method over the work of Han et al. Comparison to the work of Gong et al 2014 is not made.
The results illustrate the advantage of the hessian weighted k-means clustering criterion, and the advantages of the variable bitrate cluster encoding.  

In conclusion I would say that this is quite interesting work, although the technical novelty seems limited (but I’m not a quantisation expert).
Interestingly, the proposed techniques do not seem specific to deep conv nets, but rather generically applicable to quantisation of parameters of any model with an associated cost function for which a locally quadratic approximation can be formulated. It would be useful if the authors would discuss this point in their paper.

[Official Review · AnonReviewer3 · rating 7 · confidence 4 · 16 Dec 2016]
**Effective quantization**

The paper has two main contributions:

1) Shows that uniform quantization works well with variable length (Huffman) coding

2) Improves fixed-length quantization by proposing the Hessian-weighted k-means, as opposed to standardly used vanilla k-means. The Hessian weighting is well motivated, and it is also explained how to use an efficient approximation "for free" when using the Adam optimizer, which is quite neat. As opposed to vanilla k-means, one of the main benefits of this approach (apart from improved performance) is that no tuning on per-layer compression rates is required, as this is achieved for free.

To conclude, I like the paper: (1) is not really novel but it doesn't seem other papers have done this before so it's nice to know it works well, and (2) is quite neat and also works well. The paper is easy to follow, results are good. My only complaint is that it's a bit too long.

Minor note - I still don't understand the parts about storing "additional bits for each binary codeword for layer indication" when doing layer-by-layer quantization. What's the problem of just having an array of quantized weight values for each layer, i.e. q[0][:] would store all quantized weights for layer 0, q[1][:] for layer 1 etc, and for each layer you would have the codebook. So the only overhead over joint quantization is storing the codebook for each layer, which is insignificant. I don't understand the "additional bit" part. But anyway, this is really not a important as I don't think it affects the paper at all, just authors might want to additionally clarify this point (maybe I'm missing something obvious, but if I am then it's likely some other people will as well).

[Official Review · AnonReviewer4 · rating 7 · confidence 3 · 04 Jan 2017 (modified: 12 Jan 2017)]
**Review for "Towards the Limit of Network Quantization"**

This paper proposes a network quantization method for compressing the parameters of neural networks, therefore, compressing the amount of storage needed for the parameters.  The authors assume that the network is already pruned and aim for compressing the non-pruned parameters. The problem of network compression is a well-motivated problem and of interest to the ICLR community. 

The main drawback of the paper is its novelty. The paper is heavily built on the results of Han 2015 and only marginally extends Han 2015 to overcome its drawbacks. It should be noted that the proposed method in this paper has not been proposed before. 

The paper is well-structured and easy to follow. Although it heavily builds on Han 2015, it is still much longer than Han 2015. I believe that there is still some redundancy in the paper. The experiments section starts on Page 12 whereas for Han 2015 the experiments start on page 5. Therefore, I believe much of the introductory text is redundant and can be efficiently cut. 

Experimental results in the paper show good compression performance compared to Han 2015 while losing very little accuracy. Can the authors mention why there is no comparison with Hang 2015 on ResNet in Table 1?

Some comments:
1) It is not clear whether the procedure depicted in figure 1 is the authors’ contribution or has been in the literature.
2) In section 4.1 the authors approximate the hessian matrix with a diagonal matrix. Can the authors please explain how this approximation affects the final compression? Also how much does one lose by making such an approximation?

minor typos (These are for the revised version of the paper):
1) Page 2, Parag 3, 3rd line from the end: fined-tuned -> fine-tuned
2) Page 2, one para to the end, last line: assigned for -> assigned to
3) Page 5, line 2, same as above
4) Page 8, Section 5, Line 3: explore -> explored

[Author Response · Yoojin Choi · 14 Jan 2017]
**Revision Jan-13**

Dear Reviewers,

We have uploaded a revised paper.

1) We reduced the length of the paper to 10 pages excluding references and Appendix. To this end, 

- We reduced the length of Abstract.
- We curtailed the first three sections substantially while removing Figure 1 and Figure 2.
- We removed the detailed discussion on the alternative of Hessian in Section 4.4.
- We removed the detailed discussion on the iterative ECSQ algorithm in Section 5.4 while moving some to Appendix A.2.
- We removed the experiment results for LeNet.

We will try to reduce it further next week. Please let us know your opinion. We are open to reduce it further.

2) We added Remark 1 in Section 4.1 to address the reviewer's comment on the effect of diagonal approximation:

"Remark 1. The diagonal approximation for Hessian simplifies the optimization problem as well as its solution for network quantization. This simplification however comes with some performance loss. We conjecture that the loss due to this approximation is small. The reason is that the contributions from off-diagonal terms are not always additive and their summation may end up with a small value. However, diagonal terms are all non-negative and therefore their contributions are always additive."

3) We added Remark 3 in Section 4.1 to address the reviewer's comment on the generalization of Hessian-weighted quantization methods to other models:

"Remark 3. Observe that the relation of the Hessian-weighted distortion measure to the quantization loss holds for any model for which the objective function can be approximated as a quadratic function with respect to the parameters to quantize. Hence, the quantization methods proposed in this paper to minimize the Hessian-weighted distortion measure are not specific to neural networks but are generally applicable to quantization of parameters of any model whose objective function is locally quadratic with respect to its parameters approximately."

4) We elaborated the exponential time complexity of layer-by-layer quantization in the last paragraph of Section 4.5:

"Optimizing compression ratios jointly across all layers (to maximize the overall compression ratio for all layers) requires exponential time complexity with respect to the number of layers. This is because the total number of possible combinations of compression ratios for individual layers increases exponentially as the number of layers increases."

5) We removed the description on the "additional bits" part from Section 4.5 to reduce the length of the paper. We note that it is just a minor implementation issue as the reviewer pointed out.

6) We clarified that [Han 2015] did not evaluate ResNet by adding a row for [Han 2015] in Table 1 and put "N/A" for ResNet.

7) We fixed the typos found by reviewers.

8) We removed Appendix A.1 and Appendix A.4 to reduce the length of the paper further.

Thank you again for your valuable comments and suggestions. We look forward to further comments.

Best,
Yoojin

[Author Response · Yoojin Choi · 15 Jan 2017]
**Revision Jan-14**

Dear Reviewers,

We have reduced the length of the paper further and it is now 9 pages (excluding references and Appendix). Here is a list of major changes:

- We removed equation (2) of the previous revision since it is redundant.
- We removed Section 2 of the previous revision, and added Section 3.1 instead in this revision.
- We removed Remarks 1-3 in Section 4.1 of the previous revision, and moved all of them to Appendix A.1 instead in this revision.
- We removed the second paragraph of Section 4.5 in the previous revision.
- We reduced the length of Remark 4 in Section 5.3 of the previous revision.

We are open to reduce it further if needed. Please let us know your opinion. Thank you.

Best,
Yoojin

[Final Decision · Program Chairs · 06 Feb 2017]
**ICLR committee final decision**

The paper proposes using quantization schemes to compress the weights of a neural network. The paper carries out a methodical study of first deriving the objective function for optimizing the quantization, and then uses various quantization schemes. Experiments show competitive performance in terms of compression and accuracy tradeoff.
 
 I am happy to go with the reviewers' recommendations to accept the paper.
 
 A minor comment:
 It is important to mention other frameworks that compress neural networks, e.g.